# Stakeholders engagement for solving mobility problems in touristic remote areas from the Baltic Sea Region

Halina Kiryluk[1], Ewa Glińska[1], Urszula Ryciuk[1]*, Kati Vierikko[2], Ewa Rollnik-Sadowska[1]

1 Faculty of Engineering Management, Bialystok University of Technology, Bialystok, Poland,
2 Environmental Policy Centre, Finnish Environment Institute (SYKE), Helsinki, Finland

* u.ryciuk@pb.ed.pl

## Abstract

Stakeholder participation is particularly important when dealing with mobility problems in touristic remote areas, in which there is a need to find sustainable solutions to increase transport accessibility. However, the literature lacks research linking the issues of establishing stakeholder groups with the most desirable level of involvement and methods ensuring involvement on the indicated level. The aim of the paper is to fill this gap on example of project dedicated to six Baltic Sea Regions. In the first stage key stakeholder groups were identified, then different methods and tools were proposed depending on levels of engagement of given group of stakeholders on solving the problems of local mobility. Two research methods were implemented–the case study and the content analysis of documents. The results of the research point to the existence of five key groups of stakeholders interested in solving transport problems of touristic remote areas: authorities, business and service operators, residents, visitors and others (like experts and NGOs). Among the five–authorities and business representatives–should be to a higher degree engaged. However, the main conclusion is that engagement local government units, when developing their own, long-term strategies for social participation, should adapt the selection of participation methods and techniques to a specific target group and the desired level of their involvement so as to include stakeholders in the co-decision processes as effectively as possible and achieve effective regional co-management.

## Introduction

Tourist destinations are facing an increasing number of challenges, so governance is essential in terms of improving cooperation between all parties involved in the process of their management–it allows them to expand their scope and make the most of the opportunities offered by the market, thereby increasing their competitiveness, whilst respecting their sustainable development [1,2]. The governance of destinations is becoming an increasingly important research topic [3–5]. Recently, the social aspects of governance, such as citizen participation and

**Funding:** This research was funded by Interreg Baltic Sea Region Programme 2014-2020.

Contract no. 100#, within the project MARA - Mobility and Accessibility in Rural Areas - New approaches for developing mobility concepts in remote areas. The funders had no role in study design, data collection and analysis, decision to publish, or preparation of the manuscript.

**Competing interests:** The authors have declared that no competing interests exist.

collaboration among stakeholders, have received increased attention [5,6]. Over the past twenty-five years there has been a growing body of tourism research on stakeholder engagement in successful tourism planning and development [7]. Literature on tourism offers the notion of "collaborative governance" of a tourism destination [8]. This indicates that the involvement of different stakeholder groups is an important element of effective management of a tourism destination [9] and a key factor of sustainable tourism development [10–13].

Bramwell and Lane [14] draw attention to the increased importance of involving a wide range of stakeholders in the planning and management of sustainable tourism, noting that this reflects the transition to more decentralized and more inclusive forms of tourism management. Involving stakeholders in the process of managing a tourist destination is particularly important when dealing with touristic remote areas, in which there is a need to find common solutions to increase communication accessibility of a given territory and meet the expectations of many stakeholders. As Hopkins [15] observes, research on sustainable tourism so far has often neglected the issue of tourism mobility. Therefore, there arises a need for studies that can contribute to and potentially accelerate transitions towards tourism-transport sustainability.

Searching for sustainable transport solutions to improve accessibility and mobility in remote regions is an important challenge today, which will not only increase those regions competitiveness, but also reduce the negative impact of transport on the environment (e.g. through the development of environmentally friendly means of transport, including e-mobility solutions, which will reduce the emission of pollutants or reduce the consumption of non-renewable resources). Sustainable transport requires long-term and integrated actions that take into account a broad perspective in the search for sustainable solutions. This perspective allows for broad stakeholder involvement, both in the process of sustainable transport planning and in supporting decisions regarding various transport initiatives [16,17].

The issue of sustainable transport is more and more popular in scientific research [15,18,19]. A broad review of research on sustainable transport, identifying contemporary topics, knowledge gaps and new research areas was made by Zhao et al. [20]. These studies say that one of the crucial topics of research in this area today is stakeholder engagement.

However, the literature on the subject lacks scientific research connecting the issues of determining the main groups of stakeholders that should be involved in solving the problems of local mobility with the most desirable level of their involvement, as well as with the selection of methods ensuring this involvement at the indicated level.

The aim of the paper is to identify methods of stakeholder engagement for solving mobility problems on the example of six different touristic remote areas located in the countries of the Baltic Sea Region. An additional objective is identification of differences in the selection of these techniques, depending on the specifics of a given stakeholder group and the level of its engagement. Practical aims of the research are to raise the awareness of local government units about the key stakeholder groups and their role in the process of solving the problems of mobility and accessibility of tourism areas as well as delivering of guidelines for working out participation strategies.

In order to achieve the intended purpose of the article, three research questions were formulated:

1. What are the main important groups of stakeholders from the point of view of the possibility of implementing project's solutions;

2. What is the level of engagement of given group of stakeholders on solving the problems of local mobility;

3. What methods of stakeholder engagement for solving mobility problems are the most adequate in relation to the identified groups and desirable level of their involvement?

Answers to these research questions were sought through case study analysis and the use of empirical content analysis of documents outlined as "Stakeholder involvement strategy" prepared for six tourist destinations of the Baltic Sea Region: Vidzeme (Latvia), Birštonas and Druskininkai (Lithuania); Zaonezhye, Karelia (Russia); Setesdal (Norway), Hajnowka district (Poland) and Ludwigslust-Parchim (Germany). All these regions are very attractive for tourists (most of them have facilities listed on the UNESCO World Heritage Site), but they have problems in terms of mobility and communication accessibility, both for residents and tourists. These are both remote areas affected by demographic changes as well as urban, administrative and economic centers. The analysis of these six case studies was part of the transnational project entitled MARA–"Mobility and Accessibility in Rural Areas–New Approaches for Developing Mobility Concepts in Remote Areas", financed from the Interreg Baltic Sea Region Programme 2014–2020, Priority 3 "Sustainable transport", Specific objective 3.2 "Accessibility of remote areas and areas affected by demographic change". The overall objective of the Interreg Baltic Sea Region Programme is to strengthen integrated territorial development and cooperation for a more innovative, more accessible and sustainable Baltic Sea Region. The detailed objective of the project was to indicate methods of improving transport accessibility of selected touristic remote areas from the countries of the Baltic Sea Region. The project was aimed at searching for innovative solutions to improve the mobility of residents and tourists, including e-mobility. Project MARA is one of the Flagships project of the first macro-regional strategy in Europe „The European Union Strategy for the Baltic Sea Region" (EUSBSR). As a Flagship project, may serve as pilot example of desired activities.

In the article the concept of collaborative governance and stakeholder involvement methods and techniques are presented. Then materials and methods are provided followed by a discussion of the results and conclusions. The study contributes to the knowledge in several ways. Firstly, it identifies key stakeholder groups in regional stakeholder involvement strategies as well as the level of participation of a given stakeholder group in the process of executing local projects associated with solving the issue of mobility among residents and tourists. Secondly, it points that the level of engaging each group should look different, which depends on the function a given group has in a local transportation project. Thirdly, the selection of engagement techniques should depend on a previously planned, suggested level of including a given stakeholder group in the local project.

## Literature review

**The concept of collaborative governance in tourist destinations.**   Collaborative governance (CG) is defined by Ansell and Gash [21] as "a governing arrangement where one or more public agencies directly engage non-state stakeholders in a collective decision-making process that is formal, consensus-oriented, and deliberative and that aims to make or implement public policy or manage public programs or assets". Robertson [8] introduced the CG perspective to the tourism context. His conclusions emphasize cooperation, coordination, and collaboration as critical success factors for collaborative governance, and highlight the role of public managers for success. Tourism governance is coordinated participation of all stakeholders in a tourist destination with a view to achieving shared goals, based on a more effective use of resources (tangible, intangible, human etc.), thus fostering different forms of commitment, synergy and collaboration between different groups of stakeholders and fostering the sustainability thereof [1,22].

Public participation is a decision-making process that involves stakeholders in carrying out a public purpose. Stakeholder participation refers to the inclusion of various stakeholders in policy-making and decision-making processes [23]. It could be understood as a process that is part of any public project development [24]. In general, stakeholders are defined as groups, organizations or persons with professional or personal interests, which are responsible or/and either affected by or may influence a problem or project and its implementation [25,26]. Stakeholder involvement for solving mobility problems in touristic areas could be defined as involvement of anyone who has an interest and/or has the ability to affect and/or is affected by touristic areas development projects. The aim of stakeholder participation is to "enhance multilateral influence and interaction between all actors" [27]. Stakeholders could be described depending on the level of power and interest they have: "players" have high interest and high power, "context setters" have high power and low interest, "subjects" have high interest but low power and "crowds" have low interest and power [28]. Cohen et al. [24] identify three main public stakeholder groups:

- strategic agents (i.e. elected officials and investors) supervising the process;

- operating agents carrying out the process (i.e. city, region staff and project partners);

- participating stakeholders (i.e. residents, non-profits, business representatives, NGOs) providing input through a structured process.

Hermans et al. [25] mention, particularly within public projects, the roles of representatives of a group of population with certain interest and local or regional experts (including academics) with special profession or experience.

According to Cohen at el. [24], "stakeholders may be involved throughout multiple project phases, including preparing, planning, implementing, and evaluating project outcomes". An integrated approach to the process of planning sustainable transport and solving mobility problems is extremely important [29]. The studies of Schmale et al. [17] show the involvement of different stakeholder groups. Barfod [16] emphasizes that an approach that allows for the active participation of stakeholders in the evaluation process of transport initiatives can serve as a helpful and effective decision-support system for finding more sustainable solutions to transport problems.

In case of a tourist destination the following stakeholders can be enumerated: local government; local government departments with links to tourism; international, national, regional and local tourism organizations; tourism developers and entrepreneurs, tourism industry operators; investors (both local and international); non-tourism business practitioners; media and opinion leaders; service industries; and the community including local community groups, indigenous people's groups and local residents [30]. Sometimes a tourist destination is even understood as a group of actors linked by mutual relationships with specific rules, where the action of each actor influences those of others so that common objectives must be defined and attained in a coordinated way [31] (p.23). The challenge is how various interests, perspectives and behaviors of stakeholders may best be linked to capture the destination's collaborative potential to the full [32]. And here there arises a need to consider the concept of stakeholder collaboration in the process of destination management [30]. This term is defined as a form of voluntary joint actions where autonomous stakeholders engage in an interactive process, using shared rules, norms and structures, to act and decide on issues related to tourism development in the region [30,33,34]. As a tourism destination encompasses multiple, interdependent stakeholders often holding different views on tourism development [30,35], there is a need to implement methods that contribute to the development of a multi-dialogue

and an on-going involvement of all destination stakeholders, which fosters negotiation, consensus, commitment, knowledge exchange and agreement between all public and private stakeholders [1]. Therefore, the following research question is proposed:

RQ1: What are the main important groups of stakeholders from the point of view of the possibility of implementing project's solutions?

## Involving stakeholders in tourist destination management

The current research on the subject of stakeholders in tourism concerns in particular: analysis of the role of social participation in stimulating tourism development [36–38], identification and learning the attitudes of key stakeholders towards tourism development [39–44], analyses of their level of involvement in tourism development [10,13,45], factors determining the level of involvement [12,46] and conditions needed to support effective partnership [47]. Despite a growing interest in this issue and observing the need to involve local stakeholders in the development of tourism, the level of such involvement in the practice of many regions is low [13]. Landorf [48] points out that the involvement of local communities in the process of planning sustainable tourism development is often missing. This is also confirmed by Haukeland's study [40], which shows that local tourism stakeholders have little influence on final management decisions. Also, Canavan [37] stresses that cooperation between public and private sector stakeholders is sometimes limited and dysfunctional. Among other things, this results in such effects as: lack of common vision and strategic direction of tourism development in the region, high level of mutual distrust between stakeholders, and sometimes conflicts and wasted resources. An important factor inhibiting social activity is often a low level of social capital, lack of expert background and a lacking ability to react to changes [49]. Hatipoglu et al. [50] point out that some regions lack institutional structures for effective cooperation, which hinders the participation of stakeholders in the process of sustainable tourism management. That is why it is important to work towards promoting sustainable social links and trustworthy partnership between stakeholders in the field of tourism destination management [40], especially based on more intensive communication as well as exchange of information and experience [42].

As Kuźniar [51] observes, the contemporary success of tourism projects carried out in a given area of reception depends not so much on touristic values, but on the attitudes and behaviors of the entities that co-create the touristic product of the region, especially on their creativity, knowledge and experience, readiness to cooperate or the ability to establish long-term relationships. In this respect the residents of the reception area play a significant role here.

The attitudes of local communities cannot be limited only to the acceptance of actions but should be reflected in the participation in creating tourism development plans, increasing the attractiveness of the tourist offer of a given town and promoting its values outside as well as building its tourist image. Especially at the stage of defining directions and priorities of tourism development, stakeholders should aim for the highest level of participation, i.e. co-determination, while detailed tourism development plans should be subject to at least social consultation procedures [39].

Stakeholder involvement, with the widest possible use of different participation methods and tools, should take place at all stages of the tourist destination management process, and above all include joint diagnosis of development problems, planning and organizational activities, decision making and solving specific problems of tourist areas [52].

In the practice of the regions, there are several positive examples of a comprehensive approach to stakeholder involvement in the sustainable tourism management process. Research carried out by Wray [47] has shown that in the planning stage of sustainable tourism in Australia a stakeholder engagement process was used, bringing together consultation work-shops, setting up a destination planning website to accept broader community input, as well as creating Stakeholder Reference Groups and citizen's juries. Another example of stakeholder engagement is presented in the work of Lindström and Larson [13], which identifies the following phases of the process of local involvement in a tourism development project: formation of a representative project group, consulting local stakeholders and elaborating results with local stakeholders for increased community collaboration.

Effective management of tourism destination and the involvement of different stakeholder groups in this process can be supported by the development of a stakeholder involvement strategy, including good communication and the involvement of the community, which can be used in the development of sustainable tourism development plans [53].

Rowe and Frewer [54] indicate communication flow as a basis for classifying different forms of participation, e.g. one-way communication (informing) or two-way communication (active involvement). Stakeholders must be identified, characterized and structured in order to make a choice among proper participatory techniques and degrees of participation [55]. Arnstein's ladder symbolizes eight degrees of participation from, as a matter of fact, nonparticipation ("manipulation" and "therapy") throughout "information", "consultation" and "placation" representing tokenism, to the most advanced forms: "partnership", "delegated power" and "citizen control" [26,56,57]. Petkovic at al. [58] identify four levels of engagement: communication when stakeholders receive information; consultation when stakeholders "provide their views, thoughts, feedback, opinions, or experiences but without a commitment to act on them"; collaboration when stakeholders are engaged but without direct control over decisions; and coproduction when stakeholders are equal members of the project team. Sturm [59], depending on the intensity of participation, identifies: information, consultation, open dialogue, influence, co-decision and decision. Still, Stelzle and Noennig [60] distinguish:

- information–informing the public, supporting the understanding of the problem and solutions;

- consultation–including giving public feedback to the analysis and decisions;

- involvement–working together with the public during the process and giving feedback how the decision was influenced by the public;

- collaboration–working together with the public on every aspect and including public advice and recommendations into the decision to the maximum possible extent;

- empowerment–putting the final decision is the hands of the public.

  Hence, the next research question is placed:

RQ2: What is the level of engagement of given group of stakeholders on solving the problems of local mobility?

## Stakeholder involvement methods and techniques

Establishing relations between local authorities and members of a local community should take place both in one-sided forms (as well as informing about decisions made and consulting proposed solutions) and in interactive forms by initiating co-decision, enabling inhabitants to

independently define problems and indicate solutions [49]. At the same time, it is worth emphasizing that active participation of inhabitants in decision-making processes is a condition for a dynamic development of the area [61].

In order to support a practical process of stakeholder participation, targeted participatory techniques must be determined [55]. There are many techniques of stakeholder participation identified in literature [62,63]. To a different extent, those techniques cover the level of involvement process and it is difficult to find the one which assures full involvement–Table 1. Only the mixture of variety of participatory techniques enables meeting all the involvement levels.

The participation techniques can be chosen once the objectives and the degree of stakeholder involvement have been defined [65]. Currently, there is no standardized method allowing to choose the most relevant participatory technique [63,66]. The choice depends on many factors, including [65–67]:

- level of involvement,

- type of stakeholders,

- local cultural and social norms,

- past events (history of development etc.),

- intended timing of the use of the techniques within the project,

- knowledge and experience of the project manager/facilitator.

Consequently, the following research question is proposed:

RQ3: What methods of stakeholder engagement for solving mobility problems are the most adequate in relation to the identified groups and desirable level of their involvement?

**Table 1. Selected techniques of stakeholder participation in the context of involvement level.**

| Participation technique | Information | Consultation | Involvement | Collaboration | Empowerment |
|---|---|---|---|---|---|
| Newsletter | X | | | | |
| Reports | X | | | | |
| Presentations, public hearings | X | X | X | | |
| Internet webpage | X | X | | | |
| Interviews, questionnaires and surveys | X | X | | | |
| Field visit and interactions | X | X | X | | |
| Workshops | | X | X | X | |
| Participatory mapping | | | X | X | X |
| Focus group discussions | | | X | X | |
| Citizen jury | | X | X | X | X |
| Geospatial/decision support system | X | X | X | X | |
| Cognitive map | X | X | X | | |
| Role playing | | | X | X | X |
| Multicriteria analysis | | | X | X | |
| Scenario analysis | | X | X | X | X |
| Consensus conference | | X | X | X | X |
| Virtual simulation and gaming | X | X | X | | |

Source: [55,64].

## Materials and methods

The authors of the paper adapted a qualitative approach in the process of obtaining research material. Such an approach serves to deepen the understanding of new or hitherto unexplored phenomena in all their diversity and complexity [68] (p. 5). According to the methodology of science, in the case of qualitative research, great importance is attached to the context and specific cases as factors explaining the issue under study. Specific cases (their history and complexity) constitute an important context enabling the understanding of a given area [69] (p. 14). The research methodology used in this article is based on inductive reasoning, which consists in moving from a set of detailed observations to discovering regularities reflecting some degree of ordering of event data [70] (pp. 45–50).

The authors used two qualitative research methods to achieve the aim of the article–the case study analysis and the content analysis of documents. The case study method facilitates exploration of a phenomenon within its context using a variety of data sources. This ensures that the issue is not explored through one lens, but rather a variety of lenses which allows for multiple facets of the phenomenon to be revealed and understood [71]. The case study analysis is part of the mentioned above transnational project MARA–"Mobility and Accessibility in Rural Areas–New Approaches for Developing Mobility Concepts in Remote Areas". Cases included in the analysis are located in six areas of the Baltic Sea Region: Vidzeme (Latvia), Birštonas and Druskininkai (Lithuania); Zaonezhye, Karelia (Russia); Setesdal (Norway); Hajnowka district (Poland) and Ludwigslust-Parchim (Germany) (Fig 1). All these regions are very attractive to tourists, but they have problems in terms of mobility and communication accessibility, both for residents and tourists. Moreover, the common challenges of the rural areas of the Baltic Sea Region are: demographic problems (population decline/demographic change), seasonal fluctuation of tourists, expensive public transport, car dependent lifestyle, many stakeholders involved, lack of using digital solution [72].

The Vidzeme Planning Region is located in north-east of Latvia, north of the Daugava River. The southwestern part of the region features the capital of Latvia–Riga. The historic Old Town of Riga is included in the UNESCO World Cultural and Natural Heritage List. Vidzeme is home to the North Vidzeme Biosphere Reserve, which is the only biosphere in Latvia. There are two railway lines and multiple regional bus lines in the Widzeme region, but no airports (only private ones) and ports.

In Lithuania the case study concerned two resort areas: Birštonas and Druskininkai, which are very popular among inhabitants of Lithuanian largest cities Vilnius and Kaunas. They are situated on the bank of the biggest Lithuanian river, the Nemunas. Birštonas is a small spa town (with five sanatoriums) located in the central part of Lithuania. It is located in the eastern part of the regional park Nemunas Loops. Druskininkai is one of the largest touristic and holiday attraction centers located in the south of Lithuania, close to Belarussian and Polish borders. Quite big transport hubs in region is Marijampolė town, which is located 45 km far away from Birštonas and 90 km from Druskininkai. Quite big transport hubs in region is Marijampolė town, which is located 45 km far away from Birštonas and 90 km from Druskininkai. The Via Baltica route runs through the region, which is a part of European Route E67.

The focus area in the Republic of Karelia (Russia) is the Zaonezhye peninsula (includes three rural settlements: Velikaya Guba, Tolvuya and Shun'ga) and the adjacent archipelago of the Kizhi skerries (about 500 islands). Zaonezhsky peninsula is a part of the Medvezh'egorsk municipal district. The Karelia region is located in the northwest of the European part of Russia, between the White Sea and Finland. The capital of the Republic of Karelia is Petrozavodsk. Its characteristic feature is the location of the second largest lake in Europe–Onega, around which a settlement network concentrates. Many communication routes run through the lake.

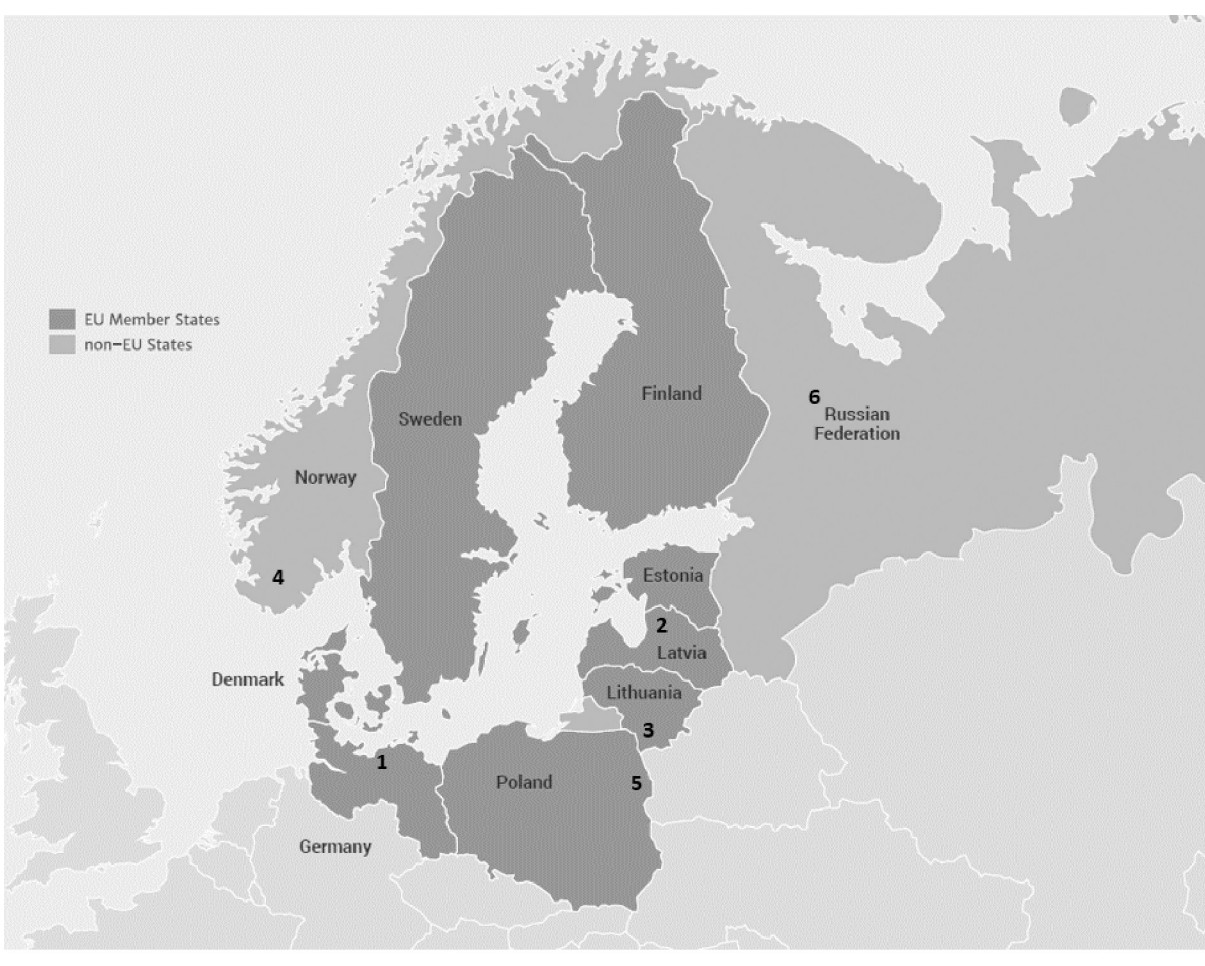

1. Ludwigslust-Parchim, Germany    3. Birštonas and Druskininkai, Lithuania    5. Hajnowka, Poland
2. Vidzeme, Latvia    4. Setesdal, Norway    6. Zaonezhye, Karelia, Russia

**Fig 1. Location map of the case studies.** Source: Own study (Map: Interreg Baltic Sea Region, interreg-baltic.eu).

Karelia is a very attractive region for tourists, both for its natural and cultural values (especially architectural monuments). The UNESCO-listed monument Kizhi island is located here. However, the region's accessibility is very limited (the road network is poorly developed) and has a very clear seasonal character (depending on weather conditions).

The Setesdal region is located in southern Norway. It is a mountainous area with a poor road system. The RV9 motorway runs through the region. The administrative center is Valle. A characteristic feature of the region is the Setesdal Valley, through which the River Otra flows. One of Setesdal's "brands" is folk music. Traditional music and dance in Setesdal (stev/stevjing) was included in the Representative List of the Intangible Cultural Heritage of Humanity UNESCO in 2019. Setesdal holds the Norwegian Sustainable Destination certificate.

The Hajnowka district is located in the south-eastern part of Podlaskie Voivodeship, which is located in the north-eastern part of Poland, near the border with Belarus. The capital of the county is the town of Hajnowka. It is part of the area known as "the Green Lungs of Poland". One of the most valuable natural sites in the world is located in Hajnowka county–the Bialowieza Primeval Forest, entered on the UNESCO World Heritage Site. The Bialowieza Forest covers 48% of the county's area. The region of the Bialowieza Primeval Forest, despite the very

high attractiveness for tourists, is a region with poor communication access (poor quality of road and rail infrastructure, low density of railways and poor offer of passenger connections, insufficient development of bicycle infrastructure, the closest airport 223 km in Warsaw) [73]. There are the Green Velo Eastern Bicycle Trail (connecting five voivodeships of Eastern Poland) and the Bialowieza Transborder Trail (linking the Bialowieza Forest on both sides of the border).

The Ludwigslust-Parchim district is located in the south-western part of Mecklenburg-Western Pomerania Federal State (also known as Mecklenburg-Vorpommern), between Hamburg and Berlin. Mecklenburg-Western Pomerania is located in the north-eastern part of Germany (border with Poland) and is one of Germany's leading tourist destinations. It is home to the largest inland lake district in Europe, with more than 1,000 lakes, including Lake Müritz. There are three parks in the federal state: "Müritz National Park", the natural park of the "Nossentin/Schwinz Heath" and "Feldberg Lake District". Ludwigslust–Parchim district has one station with a fast train connection (ICE) and good network of public buses. Next airport is in Hamburg ca. 120 km.

The basic information about the regions and the indicators of transport accessibility presents S1 Appendix.

The second research method was the content analysis of documents, namely a systematic procedure for reviewing or evaluating documents–both printed and electronic. This method focuses on the fact that data are examined and interpreted in order to obtain sense, understanding and development of empirical knowledge [74]. Document analysis may cover different categories of documents, including those in the possession of public institutions [75]. They are analyzed to determine the policies implemented by these organizations (e.g. [76,77]).

The content analysis, carried out for the purposes of the paper, covered six documents under the title "Stakeholder involvement strategy". Those documents were developed by members of the MARA project team as part of individual case studies (the above-mentioned areas located in the Baltic Sea region), included in this paper. The information was extracted from stakeholders involvement strategies elaborated for Vidzeme (Latvia), Birštonas and Druskininkai (Lithuania); Zaonezhye, Karelia (Russia); Setesdal (Norway); Hajnowka district (Poland) and Ludwigslust-Parchim (Germany) regions.

The conducted document analysis involved skimming (superficial examination), reading (thorough examination), and interpretation [74]. Research results gathered through the content analysis of those documents were analyzed with consideration for the exploratory and interpretative character of the research material [78]. In the process of analyzing data authors adopted the directed content analysis in which, during the coding process, researchers make use of both codes—formulated on the basis of the existing theory as well as the ones which they develop themselves, relying on obtained results [79]. The choice of the form of the content analysis was due to the fact that the analyzed documents were developed according to one, predetermined scheme and, therefore, contained pre-defined categories of content. However, the authors of the strategy could also propose their own options for the previously prepared scheme of the strategy document.

As it was mentioned above, strategies for engaging stakeholders in solving mobility problems were developed in the MARA project according to a predefined framework. For better presentation of applied research process including the content analysis of the strategies, below we describe the procedure on preparation of these documents, as well as their structure.

The development of strategies had the following progress: researchers from the Finnish Environment Institute (SYKE) (one of the MARA project partner) provided guidance for developing stakeholder engagement strategy. The guide prepared by SYKE contained predefined categories of content. These categories concerned such issues as: (i) a list of potential

stakeholder groups, (ii) possible levels of influence and relevance of a given group of stakeholders for solving mobility-related problems, (iii) suggested levels of their involvement in improving mobility in a given area (S2 Appendix). Pre-defined categories of content were codes used by the authors of the article in the process of analyzing the content of the "Stakeholder involvement strategy". However, no pre-defined categories of content were proposed in terms of methods/techniques for engaging stakeholders in solving mobility problems. An interactive workshop was organized by the SYKE in Hajnowka (Poland) as part of the MARA partner meeting, on September 11–12, 2019, where partners started to work on their strategy (S6 Appendix). Thirdly, the partners independently finished their regional stakeholder engagement strategies based on feedback from the SYKE.

The data analysis was made as follows. Firstly, 18 different key stakeholder groups from regional stakeholder involvement strategies were identified–all stakeholders groups mentioned in strategies were selected and counted. Secondly, in every region the impact of stakeholders on the project (on a scale from 1 to 5) and the importance of the project to stakeholders (on a scale from 1 to 5) assessed by experts were analyzed (an experts' opinion about stakeholders' influence and relevance found in regional stakeholder involvement strategies are presented in S3 and S4 Appendixes, respectively). The number of strategies in which stakeholder groups were mentioned and average scores of influence and relevance of given group for all six regions were extracted. Thirdly, for comparisons, stakeholders were grouped into five main clusters: (1) residents including diversified structure: permanent, young and senior residents, bike users; (2) authorities responsible for case introduction in the region including: local authorities, regional authorities, local spatial planners and regional spatial planners; (3) business and service operators including: transportation companies, service providers, tourist companies, regional business, local business; (4) visitors including: tourists, summer dwellers and (5) others including: researchers & experts, representatives of museums, NGOs and rescue services. Then, the influence of stakeholders and the relevance of the project in six regions were estimated–the average points from all regions and for all stakeholders in cluster were considered. In the next step, the involvement level for five main stakeholders' clusters in all six regions was presented. The involvement level presents occurrence of experts' opinions about stakeholders' level of engagement in every group. In final stage different engagement methods/ techniques mentioned in the strategies and assigned for different stakeholders were identified and counted. Experts' opinions about stakeholders' groups and level of its engagement and tools of engagement are gathered in S5 Appendix.

The authors of the paper, in carrying out research, tried to demonstrate objectivity (seeking to represent the research material fairly) and sensitivity (responding to even subtle cues to meaning) in the selection and analysis of data from documents [74]. The entire research process included three main phases consisting of more detailed research tasks (Fig 2).

## Results

Key stakeholders can significantly affect the project or are very important for its success. Without their continuous participation, the project could not be implemented. A total of 18 different key stakeholder groups from regional stakeholder involvement strategies were identified: permanent residents, young residents, bike users, local authorities, tourists, transportation operators, regional authorities, service providers, tourist companies, local business, regional business, local spatial planner, regional spatial planner, summer dwellers, NGOs and other institutions e.g. museums, researchers and experts or rescue services (Table 2). Two regions (Hajnowka and Setesdal) divided residents into subgroups based on their age, use of services or their socio-economic situation. Local authorities covered very diverse groups of actors, such

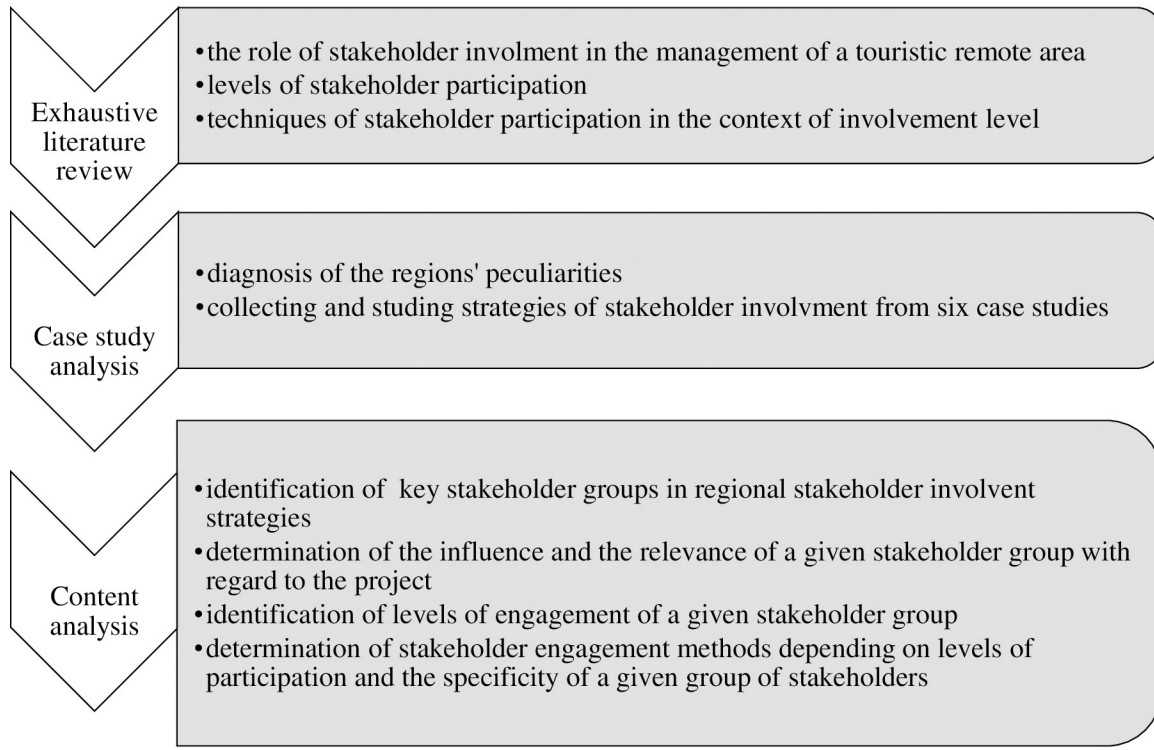

**Exhaustive literature review**
- the role of stakeholder involment in the management of a touristic remote area
- levels of stakeholder participation
- techniques of stakeholder participation in the context of involvement level

**Case study analysis**
- diagnosis of the regions' peculiarities
- collecting and studing strategies of stakeholder involvment from six case studies

**Content analysis**
- identification of key stakeholder groups in regional stakeholder involvent strategies
- determination of the influence and the relevance of a given stakeholder group with regard to the project
- identification of levels of engagement of a given stakeholder group
- determination of stakeholder engagement methods depending on levels of participation and the specificity of a given group of stakeholders

**Fig 2. Stages and tasks of the research process.** Source: Own study.

as community councils, politicians, social services, spatial planners or village leaders. Some regions identified specific stakeholders not mentioned in other regions, for example rescue services and NGOs in Zaonezhye region, Russia.

To be successful, engagement needs to be developed so that it attracts targeted audience [80–82]. In order to determine the appropriate tools for stakeholder involvement, it is important to estimate the degree to which a stakeholder has the power or capacity to influence the project or plan [83]. Although a specific stakeholder may have a low level of influence on the project, the project can still have a strong impact on this stakeholder. Therefore, it is essential to also estimate the relevance of the project to the identified stakeholder. The influence of stakeholders and the project's relevance of the stakeholders were assessed in strategies using scale from 1 –"low" to 5 –"very high" (S2 Appendix).

An average experts' opinion about stakeholders' influence and relevance found in regional stakeholder involvement strategies are presented in S3 and S4 Appendixes, respectively. Though permanent residents were most often mentioned as a key stakeholder (in all six region strategies), their influence was not considered so high (aver. 3.7) as compared to local (aver. 4.2) and regional authorities (aver. 4.8), who were scored higher, indicating that their influence on plans and goals is higher than that of local residents (Fig 3). Interestingly, transportation operators received the highest influencing scores (5) in three regions where they were considered one of key stakeholders. On the other hand, the relevance of the project to residents was scored higher (aver. 4.3), which makes sense as developing local transportation supplies and services will have a great impact on the accessibility of residents not owning a car. In three regions, residents were divided into several groups to better detect differences in influence and relevance between them.

In the next stage, for comparisons, key stakeholders from six regions were grouped into five main clusters:

**Table 2. Stakeholder groups identified in six regional stakeholder involvement strategies.**

| Stakeholder groups | Ludwigslust-Parchim, Germany | Vidzeme, Latvia | Birštonas and Druskininkai, Lithuania | Setesdal, Norway | Hajnowka, Poland | Zaonezhye Karelia, Russia |
|---|---|---|---|---|---|---|
| Residents | X | X | X | X | X | X |
| Young residents | | | | | X | |
| Residents–bike users | | | | X | X | |
| Local authorities | X | X | X | X | X | X |
| Regional authorities | X | | | X | X | X |
| Local spatial planner | | | | X | | |
| Regional spatial planner | | | | X | | |
| Transportation operators | X | | | X | X | |
| Service providers | X | X | | | | |
| Tourist companies | X | | | X | | |
| Regional business | | | | | | X |
| Local business | | | | | X | X |
| Tourists | | X | X | X | | X |
| Summer dwellers | | X | | | | |
| Researchers and experts | X | X | | | | X |
| Museums | | | | | | |
| Rescue services | | | | | | X |
| NGOs | | | | | | X |

NGO = Non-governmental organizations such as local associations or societies.

Source: Own study.

1. Residents including all kind of residents: permanent, young and older residents, infrastructure e.g. bike users.

2. Authorities including: local authorities, regional authorities, local spatial planners and regional spatial planners.

3. Business and service operators including: transportation companies, service providers, tourist companies, regional business, local business.

4. Visitors including: tourists, summer dwellers.

5. Others including: researchers and experts, museums, NGOs, rescue services.

Then, the stakeholders' influence and relevance six regions were estimated for groups–the average points from all regions and for all stakeholders in cluster were considered. The most common key stakeholder groups included permanent residents and local authorities which were mentioned in all six regions (Latvia, Vidzeme; Lithuania, Birštonas and Druskininkai; Russia, Zaonezhye, Karelia; Norway, Setesdal; Poland, Hajnowka and Germany, Ludwigslust-Parchim) (Fig 4). The influence of authorities was evaluated from 3 (in Norway, Russia and Lithuania) to 5 (in Latvia) and relevance from 3 (in Norway) to 5 (in Germany and Lithuania). This group is the most meaningful in Latvia, Germany and Poland. Residents were evaluated as having the highest influence in Latvia and Lithuania (5) and the highest relevance in Latvia (5) and Poland (4.7). Residents were assessed as the stakeholder group having the lowest influence and relevance in Russia (respectively– 3 and 2.5 average points on a 5-point scale).

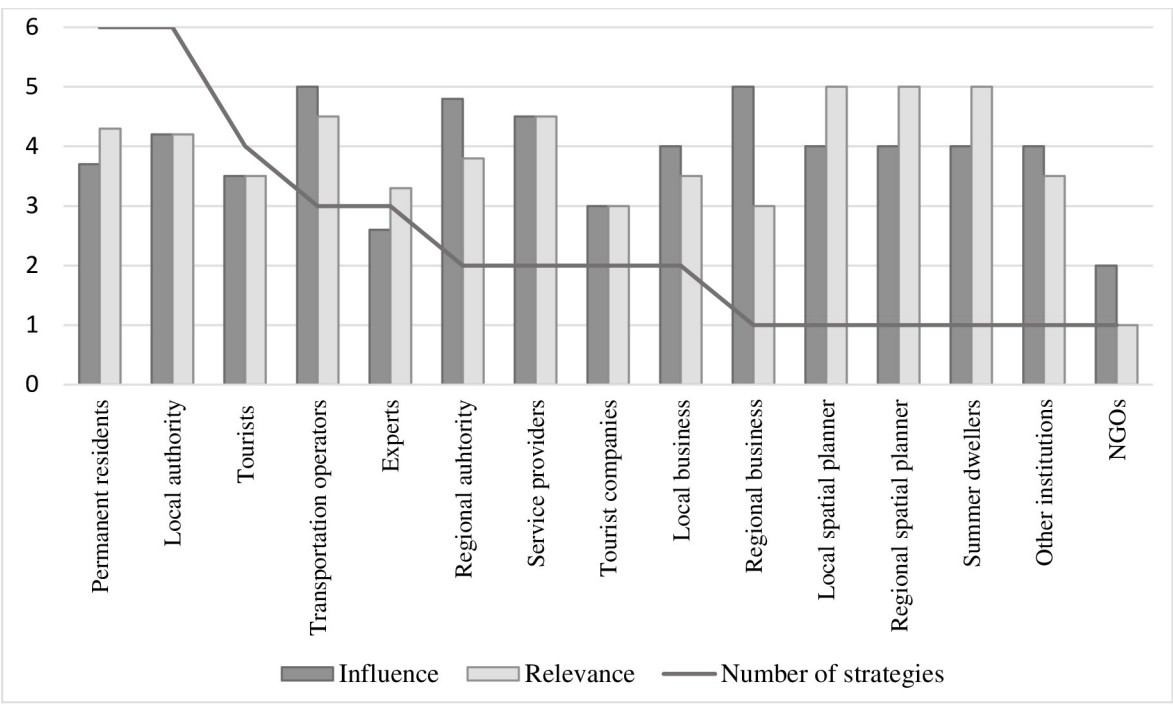

**Fig 3. Number of strategies in which stakeholder groups were mentioned and average scores of influence and relevance of given group (1 = low– 5 = very high) in six regional stakeholder involvement strategies.** Source: own study.

Visitors and business representatives were mentioned in five regions. The influence of visitors was evaluated from 3 (in Lithuania) to 5 (in Latvia) and their relevance from 3 (also in Lithuania) to 5 (in Norway and Latvia). The "visitors" group is the most important stakeholder in Latvia and Norway. Business and service operators (transportation companies, service providers, tourist companies, regional business, local business) were assessed as having the highest influence in Poland (4.5). The highest relevance at 4 points was evaluated in Latvia and Poland. Researchers, experts, museums, NGOs and rescue services, categorized in the group of "others" stakeholders (identified in three regions–in Latvia, Russia and Germany) were rated as the least meaningful in all regions.

In many planning processes a number of different participation tools are used depending on the complexity, longevity of the topic, which key stakeholders are necessary to be involved in the planning process, and if there are other groups and actors that needs to be informed. The different methods and tools can be chosen depending on the level of engagement. Therefore, the engagement level for five main stakeholders' clusters in all six regions was analyzed. The engagement level was indicated in all six regions for key stakeholders (grouped into five main clusters) on a five-degree scale (according to Stelzle and Noennig [60]), from information to empowerment (Fig 5). For some stakeholders more than one engagement level were suggested, so counted occurrence of experts' opinions about stakeholders' level of engagement in every group is presented (see S5 Appendix).

Putting the final decision in the hands of the stakeholder group was indicated in Poland (Hajnowka Region), Lithuania (Birštonas and Druskininkai Region), Russia (Karelia Region) and Norway (Setesdal Region). In three of the mentioned cases the suggested stakeholder group with the highest level of engagement was "authorities", in the fourth–researchers & experts, museums, NGO, rescue services. Collaboration is the level of engagement

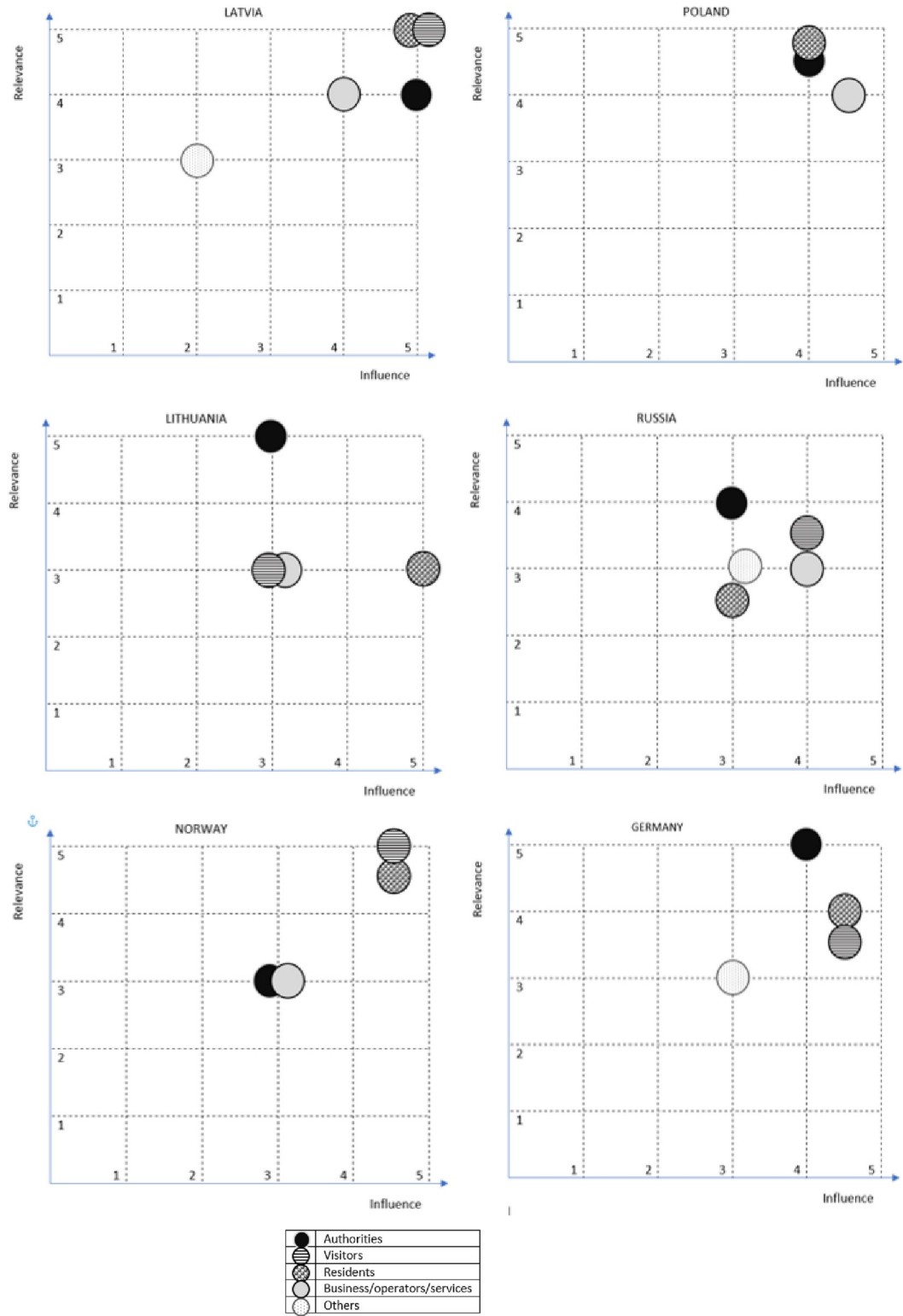

**Fig 4. Influence of stakeholders and the relevance of the project in six regions.** Source: Own study.

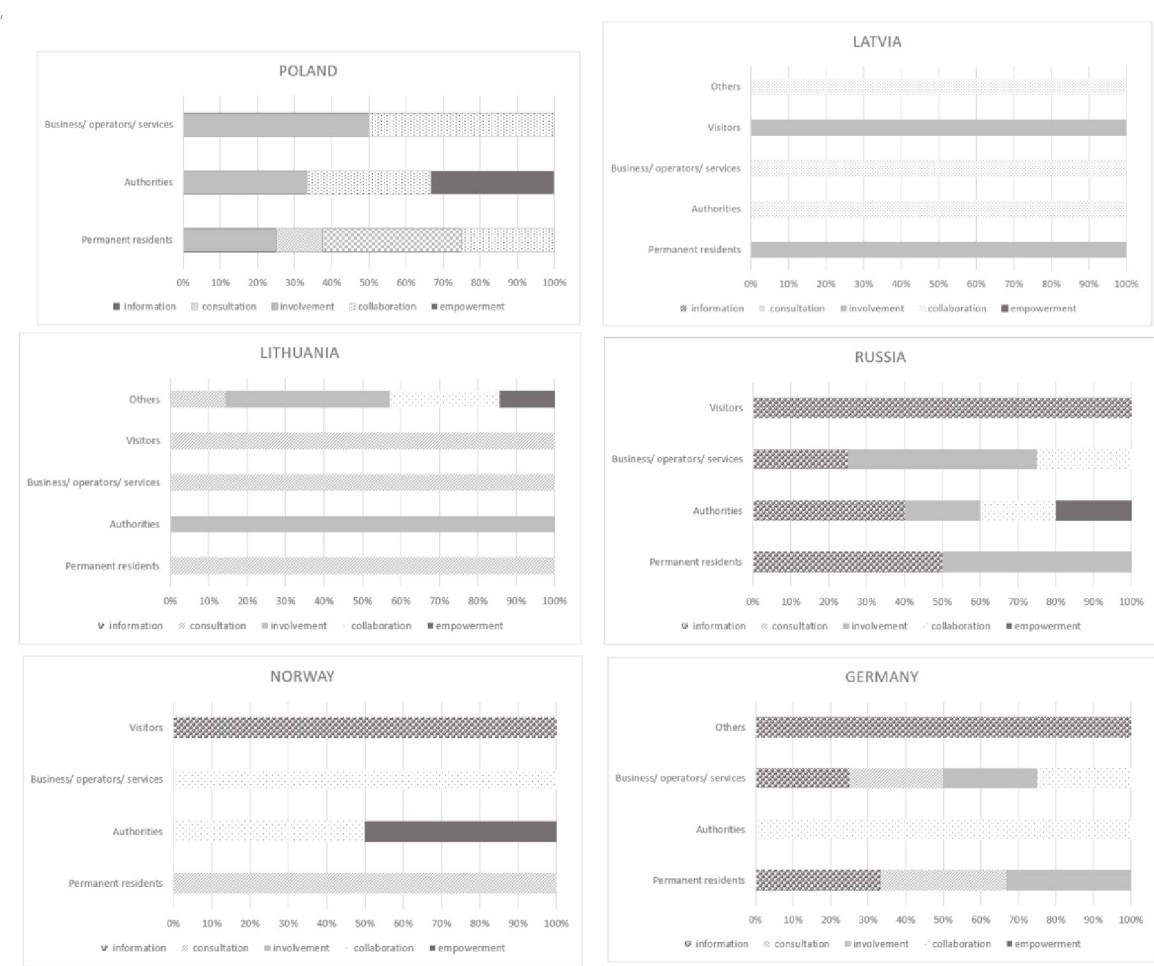

**Fig 5. Engagement levels for key stakeholder clusters in six regions.** Source: Own study.

recommended in all six regions mainly for authorities and business and service operators (in Germany, Norway, Russia, Latvia and Poland) but also residents (in Poland) and others (in Lithuania). Involvement is the level of engagement indicated in almost all regions, besides Norway, and for all stakeholder groups. However, that engagement level was the most important for the stakeholders from Poland and Russia. The groups of stakeholders who appreciated the involvement the most were residents, authorities, business and service operators. Consultation was mentioned especially for almost all groups of stakeholders in Lithuania and in four regions for residents (Poland, Lithuania, Norway and Germany). Informing stakeholders to support the understanding of the problem and solutions was recommended especially in Russia and Germany (in both cases for four groups of stakeholders).

In final stage different engagement methods/techniques mentioned in the strategies and assigned for different stakeholders were identified. Experts' opinions about stakeholders' groups and level of its engagement and tools of engagement are gathered in S5 Appendix. A total of 25 different engagement methods/techniques were mentioned in the strategies (Fig 6). Meetings, directed informing (e.g. letters) and workshops were three most common participation tools especially among residents, authorities and transportation operators. Very often it was mentioned that residents in the region need to be informed about the process of setting up a new service or improve an existing one. It was commonly highlighted that residents'

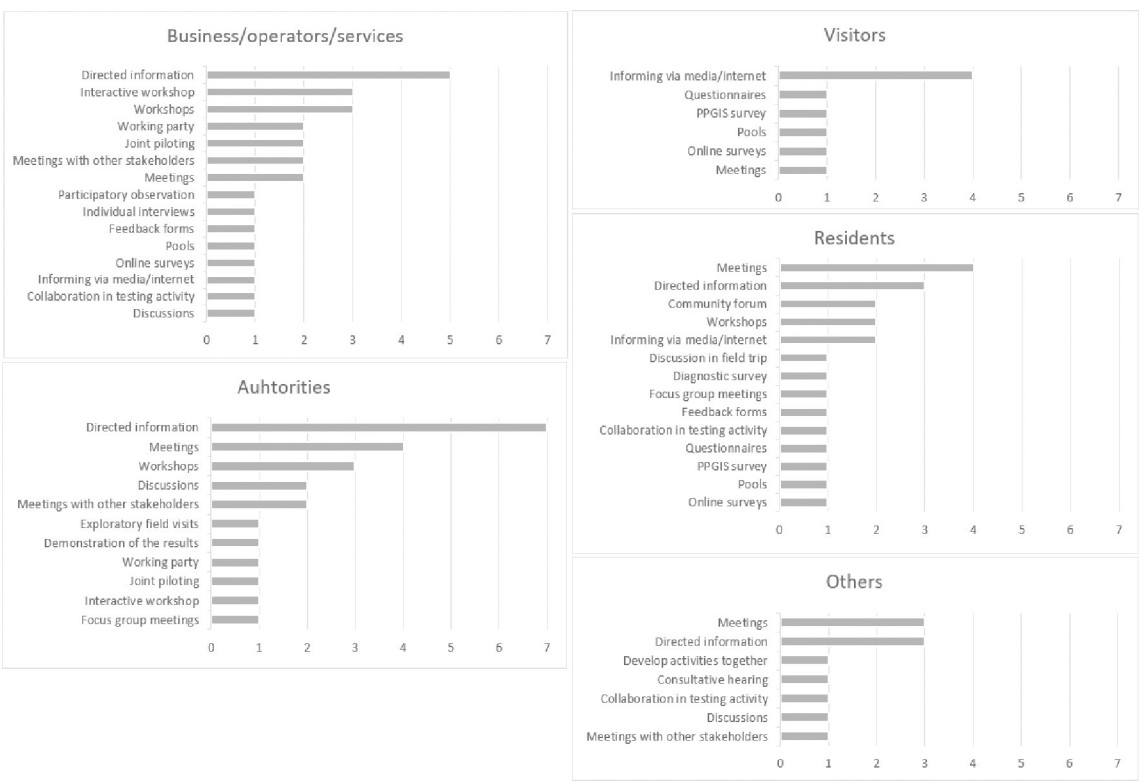

**Fig 6. The most popular engagement methods for key stakeholder clusters.** Source: Own study.

feedback and ideas need to be considered by giving them the possibility to do so. In addition, in one case vulnerable groups received careful attention and importance to collect information about mobility behaviors of socially vulnerable persons in order to understand their mobility needs. Their engagement was planned together with authorities in social services. Very often workshops with other stakeholders were mentioned among authorities, businesses and experts. Experts were invited to workshops with local/regional residents, service providers and companies or special hearings were organized. Online tools and public participation GIS tools [84] that allow for mapping specific locations or draw mobility routes on internet-based maps were mentioned in three strategies, and especially engaging residents or tourists. Many tools (8 in total) such as participatory observations, inviting stakeholders actively, developing or testing pilot or field visits were rarely mentioned.

## Discussion

Stakeholder involvement for solving mobility problems in touristic areas means engagement of anyone who has an interest and/or ability to affect and/or is affected by touristic areas development projects [25,26]. The application of this concept is of particular importance in the areas that constitute the World Heritage Sites (WHS), similarly to almost all the areas analyzed in this article. As Landorf [48] points out, these areas, in order to maintain their importance, should implement the concept of sustainable tourism, which is based on two key principles: long-term and holistic planning process and multi-stakeholder participation in this process. The analysis carried out in the paper allowed to identify and classify the key stakeholder groups and assess their impact on the project realization. It also made possible to identify and

categorize relationships between stakeholders and the methods and techniques of participation used at specific levels of involvement.

First, with regard to RQ1 the analysis of six strategies aimed at involving stakeholders in the process of solving local mobility problems showed that there are five key groups of stakeholders in such projects. These are: authorities (local and regional), residents, visitors, business/operators/services and others (including researchers and experts, museums, NGOs, rescue services).

The participation of local authorities as well as regional authorities seems crucial in understanding policy-makers and sharing information to the public. The coordinating role of local and regional governments in solving problems of tourism areas was underlined also by Saito and Ruhanen [30] and Cohen et al. [24]. Business, operators and services companies are responsible for the executive aspect related to the implementation of the project to improve mobility. The involvement of residents and visitors, as subsequent users of touristic objects, devices or networks, in the decision-making process increases the chance of creating an investment more fully suiting their needs. A necessary condition is a strong conviction of the local community of the need for a new investment. Our research confirms that local residents and visitors, being the main beneficiaries of projects aimed at improving local mobility, also make a significant contribution to those projects realization.

Non-governmental organizations include social, civic and voluntary organizations that may be interested in the project. The so-called third sector organizations are as well an important partner. These are organizations that know best the specificity of the problems they deal with (see Cohen et al. [24]).

Authorities are the most important group, recalled in six analyzed regions. The importance and the relevance of authorities in the regions are the highest (respectively 4.17 and 4.40 on a 5-point scale, Fig 7). The average relevance of business, service operators and authorities is 4.3 and the influence is 3.80. The average relevance of residents is 3.67 and the influence is 4.22. Less frequently selected groups are "visitors" and "others"–the average relevance and influence in the regions is 3.60 and 3.40 for the former group and 3.0 for the latter.

The results of the analysis included in this paper confirmed to a great extent the results of the research conducted by Saito & Ruhanen [30]. These authors emphasized the significance of such groups of stakeholders in tourist destination management as: local government; local government departments with links to tourism; tourism developers and entrepreneurs, tourism industry operators; service industries; and the community. Our research additionally indicates the "visitors" group, whose influence on the issue of solving the local mobility issue was evaluated quite high, similarly to its relevance. Among the five identified stakeholder groups, one can be qualified as "strategic agent" according to the concept presented by Cohen et al. [24]. These are local and regional authorities who are responsible for supervising the process of solving problems associated with transport accessibility of the analyzed touristic areas and business/operators/services representatives, belong to the group "operating agents carrying out the process". The remaining three (residents, visitors, and others) can be qualifies to the category "participating stakeholders". Taking into consideration the concept of Eidt et al. [28], who divide specific groups of stakeholders with regard to their level of power and interest, it should be observed that all five, which were indicated in the strategies of the analyzed areas, can be categorized as "players". They are characterized by relatively high interest and high power with regard to the project associated with improving mobility in the analyzed areas.

With regard to RQ2 the levels of engagement of given group of stakeholders on solving the problems of local mobility were identified. The analysis of the content of strategies for involving stakeholders in improving the mobility issue in the six touristic areas has shown that authorities and business/operators/services representatives, being stakeholder groups with a

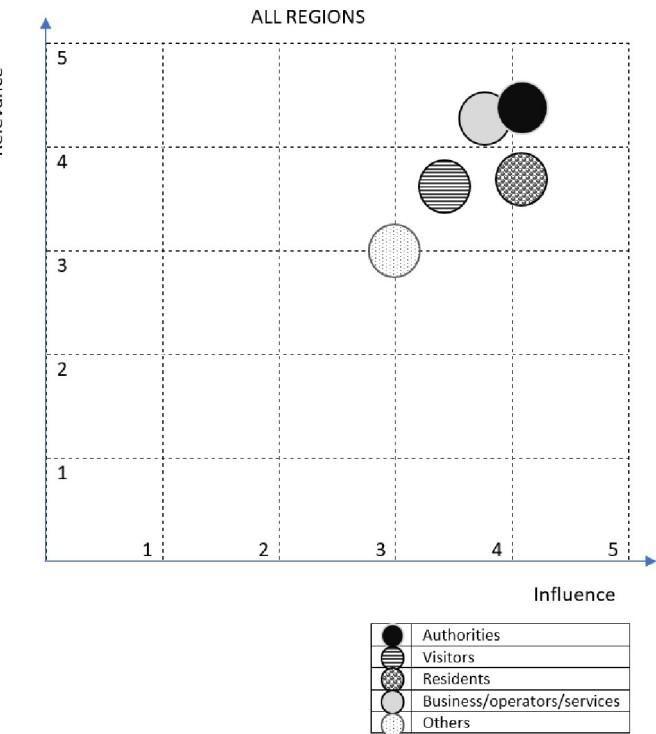

**Fig 7. Average measures of the importance and relevance of individual stakeholder groups in addressing local mobility issues.** Source: Own study.

"strategic and operating agents" status, should be to a higher degree engaged in the projects aimed at improving mobility of the analyzed touristic areas (Fig 8). In the analyzed strategies

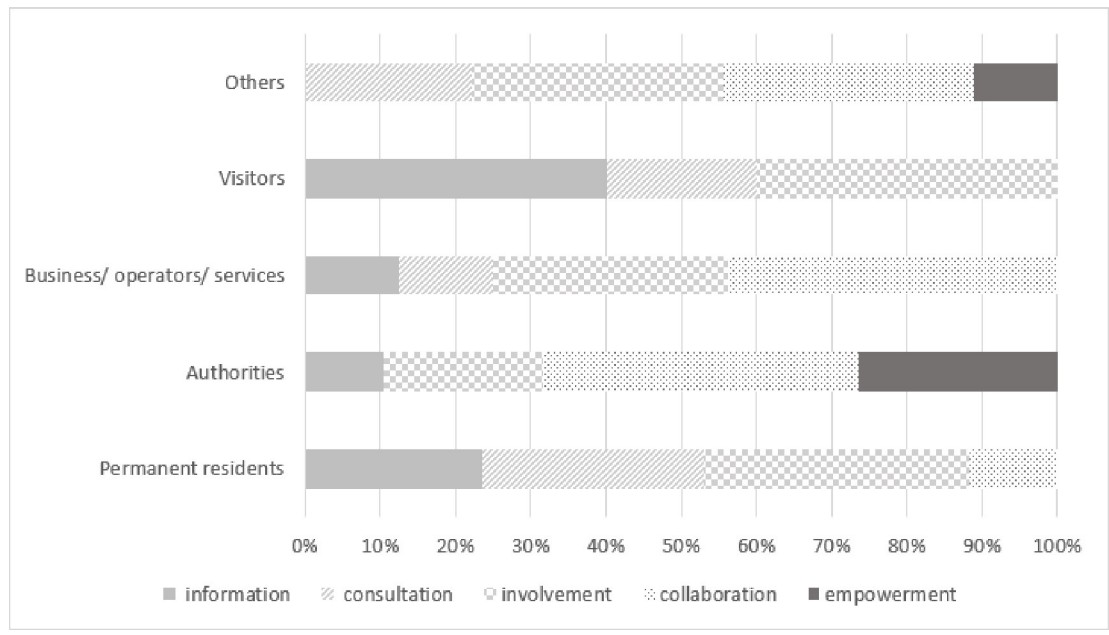

**Fig 8. Engagement levels for key stakeholder clusters in six regions.** Source: Own study.

the authors show that the contribution of authorities should most often take on the form of "collaboration" and "empowerment", and less frequently "involvement" and "information". Business/operators/services representatives should be mainly engaged through "collaboration" and "involvement", less frequently through "consultations" and "informing". Residents, constituting one of stakeholder groups with a "participating stakeholders" status, should be primarily involved at a level of "consultations" and "involvement", less frequently "information", and the least often "collaboration". For visitors, the most frequently used level of engagement should be "information" and "involvement", less often "consultation". An interesting aspect is presented by the results of the suggested engagement level among stakeholders belonging to the "others" category. Here, a special emphasis was put on "involvement" and "collaboration", but a considerably significant role was ascribed to "empowerment". This highest level of involvement is supposed to especially encompass such subcategories "others" as: researchers, experts, museums, NGOs, rescue services representatives.

The research carried out for the purposes of this article allowed to determine what methods of stakeholder engagement for solving mobility problems are the most adequate in relation to the identified groups and desirable level of their involvement (RQ3).

In transportation planning, different participation methods/techniques can be used depending on the complexity, longevity of the topic and which key stakeholders are necessary to be involved in the planning process [80]. Different participation tools may be used and selecting the most effective set of tools for engagement is crucial for the success of the whole process [81]. Stakeholder engagement is often time-consuming and needs financial effort from the organizing institution. Advance, careful planning of engagement may save time and enhance the acceptability of the project. As part of regional stakeholder involvement strategies, each partner determined the most suitable engagement method for key stakeholders. Fig 9 illustrates the methods/techniques of engaging stakeholders in the process of solving mobility issues taken from the strategy of touristic areas, as suggested for five key stakeholders groups. At the same time, the analysis incorporated the proposed level of involvement of these key stakeholder groups–from information to empowerment. The connection of the methods/techniques of engaging stakeholders in the process of solving mobility issues with the topics such as the degree of stakeholder involvement and type of stakeholders was also suggested by Reed [65], Luyet [66] and Bajarwan [85, p. 32].

In terms of "authorities", the most significant role is ascribed to such engagement levels as: collaboration, empowerment and involvement. In case of "residents", special stress should be put on involvement, consultations and information. Business/operators/services representatives should participate in joint execution of projects within mobility based on collaboration, involvement, consultations and information. For visitors the best mode would be: information, consultations and involvement. Engaging stakeholders belonging to the group "others" covers all involvement levels except for informing.

The main techniques allowing for the "collaboration" of authorities in the process of improving mobility in a touristic area are: exploratory field visit, discussion in the field trip, developing activities together, interactive workshop, collaboration in testing activities, joint piloting and focus group discussions. The techniques fostering the involvement of stakeholders include: meetings, also meetings with other stakeholders and a feedback form. Among consultation techniques the following are indicated: consultative hearing, online surveys, diagnostic survey, PPGIS survey, polls and individual interview. What is of interest–despite the fact that in the analyzed strategies of stakeholders there was the need to engage "authorities" and "others" at a level of "empowerment", none of the analyzed cases led to creating techniques that could ensure such a level of involvement of these stakeholder groups.

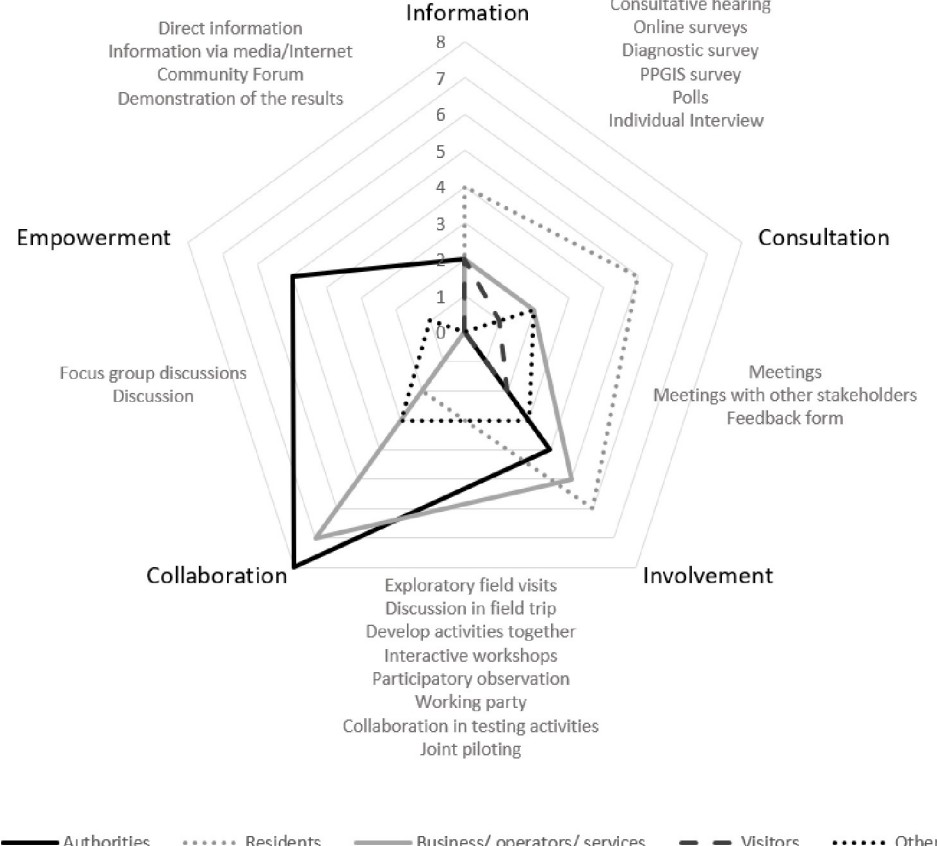

**Fig 9. Identification of stakeholder engagement methods/techniques with regard to the specifics of stakeholder groups and the level of their involvement in the process of improving the mobility problem of a given touristic area.** Source: Own study.

The research results have shown that stakeholder involvement strategies can vary greatly from region to region. Local government units, when developing their own, long-term strategies for social participation, should adapt the selection of participation methods and techniques to a specific target group and the desired level of their involvement so as to include stakeholders in the co-decision processes as effectively as possible and achieve effective regional co-management.

## Conclusions

Engaging stakeholders in managing a touristic area allows for establishing consensus and avoiding conflicts, advancing partnership and increasing responsibility as well as building trust to public institutions [49], creating and better use of the local human and social capital of touristic areas [6,86,87], as well as adds to increased quality of the project [55,88]. Stakeholders participation allows to enhance multilateral influence and interaction between all actors in a given area [49,27,89].

The article identifies key stakeholder groups in regional stakeholder involvement strategies as well as the level of participation of a given stakeholder group in the process of executing local projects associated with solving the issue of mobility among residents and tourists. The results of the research point to the existence of five groups of stakeholders interested in improving the transport situation of touristic remote areas: authorities, business and service

operators, residents, visitors and others (including experts and NGOs). The level of engaging each group should look different, which depends on the function a given group has in a local transportation project. With regard to "authorities", the most significant role is played by such involvement levels as: collaboration, empowerment and involvement. Business and services representatives should participate in joint execution of mobility-oriented projects based on collaboration, involvement, consultation and information. In case of residents, special emphasis should be placed on involvement, consultations and information. In terms of visitors, the following will be most effective: information, consultations and involvement. Engaging stakeholders belonging to the "others" group involves all involvement levels except for informing. The research indicate that engagement levels depend on stakeholders influence and relevance. Being stakeholder groups with a "strategic and operating agents" status, should be to a higher degree engaged in the projects aimed at improving mobility of the analyzed touristic areas. The selection of engagement techniques should depend on a previously planned, suggested level of including a given stakeholder group in the local project.

Selected levels of stakeholder engagement call for creating real participation (based on information flow and simple forms of co-deciding by citizens) as well as aiming at so-called ideal participation (balancing relations between public authorities and citizens). The research results show also that there grows interest of local stakeholders in co-participating in making important decisions for local development (e.g. as a result of growth in social awareness, growing social expectations and readiness for cooperation and increasing co-responsibility for decisions made together with local authorities). The conducted research shows that it in practice exists the need to raise the level of engaging local stakeholders in the execution of public undertakings with the use of a wider range of social participation tools. Citizens and social organisations that represent them should be provided with a greater possibility to co-decide in decision making.

The results of the research also indicate that, regardless of the region in which the research was conducted, there is a need to simultaneously engage many groups of stakeholders on several levels of engagement using several techniques of engaging. Different participation techniques usually depend on proposed stakeholders engagement level. However, conducted research show that even if the highest level of engagement is declared as desirable it is not related to techniques to ensure such a level of involvement. There is still need to increase the level of stakeholders engagement and simultaneously create and introduce techniques for highest level of participation.

The conducted research deepens the knowledge on the subject of stakeholder involvement in the management processes of touristic remote areas, in particular in the processes of planning mobility solutions and shaping local policy for sustainable transport. The research findings allow for a better understanding of these processes and improvement of instruments of support decision-making in the field of methods and techniques of social participation. The implemented research procedure can be helpful for other researchers in designing their own research and the study provides material for comparative analyzes.

Research may also have significant practical implications in particular for local government units. First of all, they will raise the awareness of decision-makers about the key stakeholder groups and their role in the process of solving the problems of mobility and accessibility of tourism areas and their levels of involvement. They will also allow local governments for better adaption of participation tools to specific stakeholder groups, strengthen the levels of involvement of individual groups and undertake more targeted actions in the future. The results of the research can be used as examples of good practice to develop one's own long-term participation strategies in other regions of the Baltic Sea. Such strategies will allow stakeholders to be effectively involved in the co-decision processes and to obtain acceptability of the decisions

made. The increase of stakeholders engagement will facilitate the development and the implementation of more effective and sustainable solutions of solving mobility problems in other touristic regions. It will also be conducive to the generation of new, creative ideas.

The main limitation of the carried out research may be the low representativeness of the results and a certain dose of subjectivism, both in the assessments of experts and in the interpretation of the results by the authors of the paper. Due to the nature of qualitative research, they are more exploratory than conclusive, as it is important to take into account individual contexts and specific cases. The authors realize that at each stage of the stakeholder analysis, there may have been some sources of bias [85], resulting both from the formulated research questions, the choice of research areas, and the researchers' approach. A potential source of bias could be the fact that the researchers were an active element of the study and contributed to the "Stakeholder involvement strategy" in Hajnowka District. However, all the regional strategies analyzed were developed according to uniform guidelines. Potential sources of bias could also appear in the analysis of data and the interpretation of research results, which depends not only on research methods and techniques, but also on the cognitive potential of researchers. The authors tried to reduce subjectivism in carrying out research by demonstrating objectivity (seeking to represent the research material fairly) and sensitivity (responding to even subtle cues to meaning) [74]. Thus, they strived to ensure the highest possible quality and the highest value of research.

The research was conducted in six regions of different part of Europe. The problem that connects them is the problem of mobility in remote areas. A limitation of the conducted research may be differences in the analysed regions resulting from differences in the economic development of individual countries (and therefore, regardless of needs, differences in the perceived possibilities of applying specific solutions to improve mobility in the regions); the differences in the maturity of the perception of the importance of public opinion in decision making processes in the Nordic countries in comparison to, for example, the countries of Eastern Europe. Future directions of research could take this specificity into account. It is also planned to verify, during MARA project realization, the application of the stakeholder involvement strategy and its effects in practice. The research findings may also be a starting point for future quantitative research for the authors of the paper.

## Supporting information

**S1 Appendix.**
(DOCX)

**S2 Appendix.**
(DOCX)

**S3 Appendix.**
(DOCX)

**S4 Appendix.**
(DOCX)

**S5 Appendix.**
(DOCX)

**S6 Appendix.**
(DOCX)

## Acknowledgments

Special thanks go to the leader and partners of the MARA project, as well as to the project coordinator at the Bialystok University of Technology—prof. Elzbieta Szymanska.

## Author Contributions

**Conceptualization:** Halina Kiryluk, Ewa Glińska, Urszula Ryciuk, Ewa Rollnik-Sadowska.

**Data curation:** Halina Kiryluk, Ewa Glińska, Urszula Ryciuk.

**Formal analysis:** Halina Kiryluk, Ewa Glińska, Urszula Ryciuk, Kati Vierikko, Ewa Rollnik-Sadowska.

**Investigation:** Urszula Ryciuk, Ewa Rollnik-Sadowska.

**Methodology:** Halina Kiryluk, Ewa Glińska, Urszula Ryciuk, Kati Vierikko, Ewa Rollnik-Sadowska.

**Supervision:** Ewa Glińska.

**Validation:** Halina Kiryluk, Ewa Glińska, Urszula Ryciuk, Kati Vierikko, Ewa Rollnik-Sadowska.

**Visualization:** Ewa Glińska, Urszula Ryciuk, Kati Vierikko.

**Writing – original draft:** Halina Kiryluk, Ewa Glińska, Urszula Ryciuk, Kati Vierikko, Ewa Rollnik-Sadowska.

**Writing – review & editing:** Halina Kiryluk, Ewa Glińska, Urszula Ryciuk, Kati Vierikko, Ewa Rollnik-Sadowska.

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
