## [Decision Letter · Decision Letter 0]

14 Jan 2021

PONE-D-20-37799

Stakeholders engagement for solving mobility problems in touristic remote areas from the Baltic Sea Region

PLOS ONE

Dear Dr. Ryciuk,

Thank you for submitting your manuscript to PLOS ONE. After careful consideration, we feel that it has merit but does not fully meet PLOS ONE’s publication criteria as it currently stands. Therefore, we invite you to submit a revised version of the manuscript that addresses the points raised during the review process.

We look forward to receiving your revised manuscript.

Kind regards,

Fernando Almeida-García, Ph.D

Academic Editor

PLOS ONE

Additional Editor Comments:

Thank you for submitting your manuscript to PLOS ONE. After careful consideration, we feel that it has merit but does not fully meet PLOS ONE’s publication criteria as it currently stands. Therefore, we invite you to submit a revised version of the manuscript that addresses the points raised during the review process. In particular, the reviewers note that additional clarity is needed concerning some aspects of the model and included variables, as well as further discussion of implications and limitations of the research.

Journal Requirements:

Reviewers' comments:

Reviewer's Responses to Questions

**Comments to the Author**

1. Is the manuscript technically sound, and do the data support the conclusions?

Reviewer #1: No

Reviewer #2: Yes

2. Has the statistical analysis been performed appropriately and rigorously? 

Reviewer #1: No

Reviewer #2: Yes

3. Have the authors made all data underlying the findings in their manuscript fully available?

Reviewer #1: No

Reviewer #2: Yes

4. Is the manuscript presented in an intelligible fashion and written in standard English?

Reviewer #1: Yes

Reviewer #2: Yes

5. Review Comments to the Author

Reviewer #1: Interesting article with an additional application dimension. The use of case studies in the implementation of the research problem and investigations in various locations in terms of the level of socio-economic development is worth emphasis. However the text of the article needs to be supplemented. The lack of a more precise method of obtaining the source material on the basis of which further analyzes were carried out is particularly visible. Similarly, there is no information on the stakeholder population covered by the research and no statistical reliability of the source material obtained. Were theese investigated stakeholders populations statistically representative ? It seems necessary to supplement the text of the article with basic data on the number of stakeholders participating in the research, taking into account their origin, type of stakeholders, and statistical credibility of the research sample used.

It is also advisable to define the share of individual stakeholder groups subject to the research in the total population of the relevant stakeholder population.

It also seems justified to supplement the text with basic indicators of the level of economic development of individual areas of research implementation and indicators of transport accessibility of the areas in which the research locations are located. The above-mentioned additions should be useful in better understanding and clarifying of stakeholders perception of solving transport problems in choosen touristic remote areas.

Reviewer #2: The manuscript present recent tourism studies not only concerning stakeholders engagement but also highlight the issue of mobility in remote regions.

When its regards the statistical analisys, its noted that the investigation is part of a major research that has beeing developed in the Baltic Region. More additional informational about how the research was conducted and data collection may be a plus.

The findings are clearly portraited in a standard academic way, with a proper english, well presenting all the information that regards the issues of stakeholders and mobility across the region, advancing theoretical concepts recently studied on tourism literature.

6. PLOS authors have the option to publish the peer review history of their article (what does this mean?). If published, this will include your full peer review and any attached files.

Reviewer #1: No

Reviewer #2: **Yes: **Victor Hugo da Silva

---

## [Author Response · Author response to Decision Letter 0]

17 Feb 2021

The authors wish to thank the editors and the reviewers for their time and effort in reviewing our manuscript. We believe the changes listed have addressed the points raised, making the article suitable for publication.

---

## [Editor Report · Decision Letter 1]

24 Feb 2021

PONE-D-20-37799R1

Stakeholders engagement for solving mobility problems in touristic remote areas from the Baltic Sea Region

PLOS ONE

Dear Dr. Ryciuk,

Thank you for submitting your manuscript to PLOS ONE. After careful consideration, we feel that it has merit but does not fully meet PLOS ONE’s publication criteria as it currently stands. Therefore, we invite you to submit a revised version of the manuscript that addresses the points raised during the review process.

The 11 points proposed for improvement are important and must be answered, especially those related to the methodology, since it is very poorly explained. The connection of the research questions is important and must be clearly demonstrated.

We look forward to receiving your revised manuscript.

Kind regards,

Fernando Almeida-García, Ph.D

Academic Editor

PLOS ONE

Journal Requirements:

**Additional Editor Comments:**

In general, we would expect qualitative studies to include the following:

**1)** Defined objectives or research questions. It is important to include research questions that are some of the contributions. These questions should be addressed in the discussion or in conclusions.

**2)** Description of the sampling strategy, including rationale for the recruitment method, participant inclusion / exclusion criteria and the number of participants recruited; Reference authors used to create questionnaires for interviews, pilot testing and validation of interview questionnaires.

**3)** Detailed reporting of the data collection procedures; data collection team, dates, problems, etc.

**4)** Data analysis procedures described in sufficient detail to enable replication;

**5)** Content analysis, categories used in the analysis, explanation of the content analysis process and categories, justification of the method.

**6**) A discussion of potential sources of bias;

**7**) A discussion of limitations and practical implications.

**8)** Identification of the case studies, justification of the cases and location map of the case studies.

**9)** Transcription of the stakeholder interviews (model of interviews in annex and link to the transcription of the interviews)

**10) **Justification of the identification process of the five stakeholders

**11) **Results linked to specific statements from stakeholder interviews

In the next submission of the article, I would like you to include an individual file with a document in which each of the points is answered.

Best regards,

Fernando Almeida

---

## [Author Response · Author response to Decision Letter 1]

26 Mar 2021

The authors wish to thank the editors and the reviewers for their time and effort in reviewing our manuscript. We believe the changes listed have addressed the points raised, making the article suitable for publication.

---

## [Editor Report · Decision Letter 2]

7 Apr 2021

PONE-D-20-37799R2

Stakeholders engagement for solving mobility problems in touristic remote areas from the Baltic Sea Region

PLOS ONE

Dear Dr. Urszula Ryciuk, 

Thank you for submitting your manuscript to PLOS ONE. After careful consideration, we feel that it has merit but does not fully meet PLOS ONE’s publication criteria as it currently stands. Therefore, we invite you to submit a revised version of the manuscript that addresses the points raised during the review process.

Dear authors,

Take the time that is necessary to answer all the questions appropriately. The review of the whole indicated aspects has not been satisfactory, so they should carefully review your research. I would like you answer the mentioned points in the next revision.

1) Regarding the research problems, they are not clearly debated in the discussion section. This section needs more depth of analysis. They should also explain why these three research questions have been chosen.

2) "Reference authors used to create questionnaires for interviews, pilot testing and validation of interview questionnaires". This question is not answered, these aspects are important to give credibility to your research. 

3) They must answer all aspects requested in section 3. "Detailed reporting of the data collection procedures; data collection team, dates, problems, etc." Qualitative research studies should be reported in accordance to the Consolidated criteria for reporting qualitative research (COREQ) checklist or Standards for reporting qualitative research (SRQR) checklist. Further reporting guidelines can be found in the Equator Network's Guidelines for reporting qualitative research.

4) Explain what methods of analysis have been used to achieve the results. Not only the choice of the stakeholders, but the analysis methods used to extract information from the questionnaires.

5) Content analysis, categories used in the analysis, explanation of the content analysis process and categories, justification of the method. Where are the analysis categories of your questionnaires and research? They need a read on the categories in qualitative analysis.

6)  A discussion of potential sources of bias. Some mention is made in the limitations.

7) In relation to the research title and research questions, there should be some more practical implication.

9) Transcription of the stakeholder interviews (model of interviews in annex and link to the transcription of the interviews). Interview transcripts should be included in an attached file. Some of the stakeholder opinions should be integrated into the text to support the discussion, and the research questions; also identify the main stakeholder profiles in relation to the level of influence or engagement.

11) Results linked to specific statements from stakeholder interviews. There are no comments from the interviews that provide greater richness to the analysis and plausibility.

We look forward to receiving your revised manuscript.

Kind regards,

Fernando Almeida-García, Ph.D

Academic Editor

PLOS ONE

---

## [Author Response · Author response to Decision Letter 2]

21 May 2021

Dear Fernando Almeida-García thank you for their time and effort in reviewing our manuscript. We believe the changes listed have addressed the points raised, making the article suitable for publication.

---

## [Editor Report · Decision Letter 3]

27 May 2021

PONE-D-20-37799R3

Stakeholders engagement for solving mobility problems in touristic remote areas from the Baltic Sea Region

PLOS ONE

Dear Dr. Ryciuk, 

Thank you for submitting your manuscript to PLOS ONE. After careful consideration, we feel that it has merit but does not fully meet PLOS ONE’s publication criteria as it currently stands. Therefore, we invite you to submit a revised version of the manuscript that addresses the points raised during the review process.

The article has improved substantially and I think it can be published with some minor changes. You have worked on the aspect that was the most confusing, which was the methodology.

I am going to request the following minor aspects:

- Clearly explain the contribution of the research (in Introduction).

- Include as a secondary objective the practical implication of the research, so that the objectives are connected with the conclusions.

- Methodology. You mention that some workshops were held from which information was obtained. Explain the function of these workshops (number, dates, places, information gathering process, etc.), including photographs, if any, are included in the annex.

We look forward to receiving your revised manuscript.

Kind regards,

Fernando Almeida-García, Ph.D

Academic Editor

PLOS ONE
---

## [Author Response · Author response to Decision Letter 3]

28 May 2021

The authors wish to thank the editor for their time and effort in reviewing our manuscript. We believe the changes listed have addressed the points raised, making the article suitable for publication.

---

## [Editor Report · Decision Letter 4]

1 Jun 2021

Stakeholders engagement for solving mobility problems in touristic remote areas from the Baltic Sea Region

PONE-D-20-37799R4

Dear Dr. Ryciuk,

We’re pleased to inform you that your manuscript has been judged scientifically suitable for publication and will be formally accepted for publication once it meets all outstanding technical requirements.

Kind regards,

Fernando Almeida-García, Ph.D

Academic Editor

PLOS ONE

Additional Editor Comments:

The authors have resolved the doubts and problems that the article presented, I thank you for your perseverance and work, I am sure that your research has improved.

---

## [Editor Report · Acceptance letter]

15 Jun 2021

PONE-D-20-37799R4 

Stakeholders engagement for solving mobility problems in touristic remote areas from the Baltic Sea Region 

Dear Dr. Ryciuk:

I'm pleased to inform you that your manuscript has been deemed suitable for publication in PLOS ONE. Congratulations! Your manuscript is now with our production department. 

Kind regards, 

on behalf of

Dr. Fernando Almeida-García 

Academic Editor

PLOS ONE